# Effects of the Practice of Movement Representation Techniques in People Undergoing Knee and Hip Arthroplasty: A Systematic Review

**DOI:** 10.3390/sports10120198

**Published:** 2022-12-05

**Authors:** Cristóbal Riquelme-Hernández, Juan Pablo Reyes-Barría, Abner Vargas, Yaynel Gonzalez-Robaina, Rafael Zapata-Lamana, David Toloza-Ramirez, Maria Antonia Parra-Rizo, Igor Cigarroa

**Affiliations:** 1Escuela de Kinesiología, Departamento de Salud, Universidad Arturo Prat, Victoria 4720000, Chile; 2Escuela de Kinesiología, Departamento de Salud, Universidad de los Lagos, Puerto Montt 5480000, Chile; 3Clínica Resilient, Puerto Montt 5480000, Chile; 4Escuela de Educación, Universidad de Concepción, Los Ángeles 4440000, Chile; 5Exercise and Rehabilitation Sciences Institute, School of Speech Therapy, Faculty of Rehabilitation Sciences, Universidad Andres Bello, Santiago 7591538, Chile; 6Interdisciplinary Center for Neuroscience, Faculty of Medicine, Pontificia Universidad Católica de Chile, Santiago 8320000, Chile; 7Faculty of Health Sciences, Valencian International University (VIU), 46002 Valencia, Spain; 8Department of Health Psychology, Faculty of Social and Health Sciences, Campus of Elche, Miguel Hernández University (UMH), 03202 Elche, Spain; 9Escuela de Kinesiología, Facultad de Salud, Universidad Santo Tomás, Los Ángeles 4440000, Chile

**Keywords:** arthroplasty, arthrosis, movement representation techniques, motor imagery, observation of the action

## Abstract

Objective: To analyze the effects of movement representation techniques (MRT) combined with conventional physical therapy (CFT) in people undergoing knee and hip arthroplasty compared to conventional physical therapy alone in terms of results in physical and functionality variables, cognitive function, and quality of life. Methodology: the review was carried out according to the criteria of the PRISMA statement, considering studies in the electronic databases PubMed/Medline, Pubmed Central/Medline, Web of Science, EBSCO, and ScienceDirect. Results: MRT plus CFT generated therapeutic effects in some aspects of the physical variables: 100% pain (7 of 7 studies); 100% strength (5 out of 5 studies); range of motion 87.5% (7 out of 8 studies); 100% speed (1 of 1 study), functional variables: 100% gait (7 of 7 studies); functional capacity 87.5% (7 out of 8 studies); cognitive variables: 100% motor visualization ability (2 out of 2 studies); cognitive performance 100% (2 of 2 studies); and quality of life 66.6% (2 of 3 studies). When comparing its effects with conventional physical therapy, the variables that reported the greatest statistically significant changes were motor visualization ability, speed, pain, strength and gait. The most used MRT was motor imagery (MI), and the average time extension of therapies was 3.5 weeks. Conclusions: movement representation techniques combined with conventional physical therapy are an innocuous and low-cost therapeutic intervention with therapeutic effects in patients with knee arthroplasty (KA) and hip arthroplasty (HA), and this combination generates greater therapeutic effects in physical, functional, and cognitive variables than conventional physical therapy alone.

## 1. Introduction

In traumatology and orthopedic surgery, knee arthroplasty (KA) and hip arthroplasty (HA) are documented surgical treatments for osteoarthritis of the knee and hip. These interventions significantly favor the quality of life in patients, both for their effective results and cost-effectiveness [1]. Global Burden of Disease studies recently indicated that knee osteoarthritis is the fastest-growing leading health disorder and the second leading cause of disability worldwide [2]. The risk of disability associated with osteoarthritis is equal to that of cardiac diseases [3]; indeed, 65% of the KA correspond to people aged 65 years and over. In the same way, an increase of this surgery has been projected in younger people [4,5]. On the contrary, the effects prior to surgery and the immediate post-surgical deteriorate cardiopulmonary parameters and increase the risk of falls and movement disorders [6]. Moreover, preoperative, and postoperative periods have increased immobilization due to other physical, psychological, and socioeconomic comorbidities [7,8]. Regarding the physical sphere, other indicators such as pain, largely pharmacologically managed [9,10,11], and the range of motion could also be affected. Likewise, those could affect preoperative and postoperative [12,13] and muscle strength, experiencing early and late postoperative deficits [14,15], among other indicators of functionality indicated by the literature. According to the consequences of the disease and surgery, it is relevant to consider therapeutic interventions that favor the recovery of patients.

Movement representation techniques (MRT) refer to interventions that generate a mental simulation of a motor action without performing it physically [16,17]. It is a cognitive representation process that uses images of motor tasks to activate brain regions associated with motor preparation and action [18]. MRT enclose different types of interventions, including action observation therapy (AOT), mirror therapy (MT), and motor imagery (MI). In relation to the latter, MI is defined as a process by which a subject mentally simulates a movement without real execution; AOT refers to the perception of a movement produced by others, observed in its real execution, video or virtual reality, MT is an intervention that uses a mirror to create a reflection of the unaffected limb, generating visual feedback of normal movement without the pain of the affected limb of the patient [19]. Through advances in neurosciences, this type of intervention has been extended to different fields of rehabilitation; athletes [20,21], spinal cord injuries, sequelae of cerebrovascular diseases [22,23], and orthopedic surgery [24]. Interventions such as MRT could be a promising adjunctive strategy to favor the development and rehabilitation of physical health variables [25,26]. However, it remains a therapeutic tool little used by rehabilitation teams, particularly in traumatology. The scarce evidence presents heterogeneous methodologies of low methodological quality [27,28]. Thus, there is currently no consensus on the acute and chronic effects of rehabilitating people undergoing arthroplasty, so this review could be the most up-to-date evidence on the effects on physical health, cognitive functions, and quality of life of MRT intervention combined with CPT in patients with arthroplasty. In addition, a systematic review could help clarify which is the most widely used MRT intervention in research and what are the characteristics of the intervention using MRT. Along these lines, this review aims to analyze the effects of movement representation techniques combined with conventional physical therapy in people undergoing knee and hip arthroplasty compared to conventional physical therapy alone in terms of results in physical and functionality variables, cognitive function, and quality of life.

## 2. Materials and Methods

This systematic review was conducted in accordance with the standards established by the PRISMA statement [29]. The PRISMA checklist can be found in the Appendix A. The study was endorsed in the prospective international register of systematic reviews PROSPERO (Prospero Code: CRD42022313096).

### 2.1. Search Strategy for Study Identification

For this research, a systematic review of the literature was developed in the following electronic databases, in the order indicated: PubMed/Medline, Pubmed Central/Medline, Web of Science, EBSCO, and ScienceDirect. We identified with the Zotero program all the articles published in the databases that studied the effects of movement representation techniques in people undergoing knee and hip arthroplasty. For the search, keywords, MeSH, and free terms were used, on which the general syntax of the search was generated: “Arthroplasty” OR “Arthroplasty, Replacement” OR “Hemiarthroplasty” AND “motor imagery” OR “action observation” OR “grade motor imagery” OR “mirror therapy” OR “movement representation techniques”. The literature search was conducted in October 2021 with no limits for previous years and no language restrictions. The complete search strategies are presented in the Appendix A.

### 2.2. Selection of Studies

The selection of studies is presented in the PRISMA flowchart. We included randomized and non-randomized clinical trials, excluding systematic reviews, quasi-experimental studies, observational studies, editorial papers, conference proceedings, protocols, and theses. Articles selected by title, abstract and full text had to meet the conditions indicated in Table 1.

### 2.3. Data Extraction

In the first instance, duplicate articles were removed using the Zotero program. Then, three reviewers (C.R.-H., J.R.-B. and A.V.-B.) independently assessed whether the studies met the eligibility criteria by reading only titles and abstracts. Likewise, in case of discrepancy and determining an article as “uncertain” a fourth reviewer (I.C.) intervened to achieve a consensus, inclusion/exclusion criteria were determined using the PICOT format (population, intervention, comparison, outcomes, and type of study). In the following screening, the documents were reviewed in full text by one of the reviewers (I.C.), determining the number of final articles included in the research.

### 2.4. Risk of Bias Assessment Tool

We used the Cochrane Collaboration tool to assess the risk of bias in randomized clinical trials, which covers six domains of bias: (D1) selection bias, (D2) performance bias, (D3) detection bias, (D4) attrition bias, (D5) reporting bias, and overall) the overall risk of bias. Within each domain, evaluations were performed for one or more elements, assigning a judgment of high, low, or unclear risk of bias [30]. Each article was rated by one of the reviewers (I.C.), the results of each article are presented in a traffic light diagram and the synthesis of risk of bias.

### 2.5. Strategy for Data Synthesis

We synthesized the evidence from the included studies and presented relevant information in summary tables and figures. The stratification of the selected studies was represented by the PRISMA flowchart, and the rest of the information was collected as follows: the general characteristics of the articles (author, year, country, population, sex, intervention, variables and type of design); Characteristics of the interventions (frequency, treatment, extent and duration); description of the effects of the interventions (physical, functional, cognitive functions and quality of life).

## 3. Results

### 3.1. Literature Search

In the initial search, 57 scientific articles were identified, of which 15 were duplicates. After reading by titles and abstracts, 32 studies were excluded for not addressing the inclusion and exclusion criteria, leaving 10 for full-text reading. We finally selected ten articles for this review (Figure 1).

PRISMA 2020 flow diagram for new systematic reviews which included searches of databases and registers only [29].

### 3.2. General Characteristics of the Studies

A total of 10 articles were included. The years of production of the studies ranged between 2016 and 2020, and 90% of the research was of European origin. The interventions of movement representation techniques were Motor imagery, Action observation, combined techniques (MI, AOT), and augmented reality. 80% of the studies were carried out in subjects with total knee arthroplasty, and 20% in subjects with total hip arthroplasty. On the one hand, in the study variables, 80% evaluated functional capacity, 70% some manifestation of pain, 70% ROM, 60% gait, 50% strength, 30% quality of life and 20% motor visualization capacity, speed, and cognitive performance. On the other hand, referring to the methodological characteristics of the selected studies, the most used type of design was the randomized clinical trial (90%) (Table 2).

### 3.3. Risk of Bias in Articles

Risk of bias analysis revealed that the distribution of biases classified as “low risk”, or “unclear risk” was similar in domains: (D1) selection bias and (D5) reporting bias. In domains (D2) performance bias and (D3) detection bias, a low percentage of articles with “low risk” (1/10 and 3/10, respectively) and a high percentage of articles with “unclear risk” (9/10 and 5/10, respectively) were evidenced. Conversely, in domain (D5) attrition bias and overall risk of bias, high percentages of “low risk” were observed (9/10 and 10/10, respectively) (Figure 2 and Figure 3).

### 3.4. Characteristics of Interventions Using Movement Representation Techniques (MRT)

MRT interventions were carried out as: Motor imagery 60%, Action observation 20%, combined techniques (MI, AOT) 10%, and augmented reality 10%. 60% of the studies used a weekly frequency of MRT of 5 days, 30% applied three days, and 10% six days; on average, the frequency was 4.5 days a week. The shortest intervention extension was one week, and the longest was nine weeks; moreover, the average of therapies was 3.5 weeks. All interventions based on movement representation techniques were used in a complementary way to conventional physical therapy (CPT). CPT recorded four therapeutic areas: active mobilization, passive mobilization, practice of transfers and written information; 70% of the studies described the use of all four techniques respecting (Table 3).

### 3.5. Effect of Motion Representation Techniques

Table 4 describes the effects of interventions of movement representation techniques plus conventional physical therapy (CPT) compared with the control group (conventional physical therapy) in some additional placebo studies.

We found the following results. The effects of the therapeutic intervention were analyzed, and these were grouped into four variables: physical, functional, cognitive, and quality of life, which were distributed into eight subcategories. The most analyzed variable was physics, classified into subcategories: pain, ROM, strength, and speed. The most analyzed subcategory was functional capacity and ROM.

When analyzing the post-MRT effects combined with CPT, statistically significant improvements could be observed in: physical variables such as 100% pain (7 of 7 studies), 100% muscle strength (5 out of 5 studies), ROM 87.5% (7 out of 8 studies), 100% speed (1 of 1 study); functional variables: 100% gait (7 of 7 studies), functional capacity 87.5% (7 out of 8 studies); cognitive variables: 100% motor visualization ability (2 out of 2 studies), cognitive performance 100% (2 of 2 studies) and quality of life 66.6% (2 of 3 studies).

When comparing MRT combined with CPT to the control group (CPT), the findings with statistically significant improvements in decreasing order were: motor visualization capacity 100% (2 of 2 studies), speed 100% (1 of 1 study), pain 71% (5 of 7 studies), strength 60% (3 of 5 studies), gait 57% (4 of 7 studies), functional capacity 50% (4 out of 8 studies), cognitive performance 50% (1 out of 2 studies), ROM 37.5% (3 out of 8 studies), and quality of life 33% (1 out of 3 studies).

Motor visualization capacity 100% (2 of 2 studies), speed 100% (1 of 1 studies), pain 71% (5 of 7 studies), strength 60% (3 of 5 studies), and gait 57% (4 of 7 studies) were the variables that reported the greatest statistically significant changes, following the intervention of MRT plus CPT.

A study that evaluated the variable gait presented dissimilar results, the group intervened with MRT obtained periods of double and single support significantly better; however, the control group had better stride length and cadence.

One study recorded falls and near falls following arthroplasty and MRT plus CPT intervention, in which the intervention group had significantly fewer events than the control group. Two studies significantly attenuated the deterioration of strength and gait variables with respect to the participants’ condition prior to surgery (they performed the first evaluation prior to surgery). No studies reported side effects when adding MRT to conventional physical therapy.

## 4. Discussion

### 4.1. What Are the Main Results?

The results suggest that the main categories evaluated by the studies are physical, functional, cognitive, and quality of life. The most evaluated category was the physical one; the functional capacity subcategory and ROM were the most analyzed, with 80% (8 of 10 studies). In addition, the most used technique to carry out MRT was motor imagery, holding a 60% (6 out of 10 studies). Regarding the effects of MRT intervention plus CPT compared to control (conventional physical therapy). Motor visualization ability (100% 2 of 2 studies), speed (100% 1 of 1 studies), pain (71% 5 of 7 studies), strength (60% 3 of 5 studies), and gait (57% 4 of 7 studies) were the variables that reported the greatest statistically significant changes. MRTs are considered a safe and low-cost technique because no studies reported adverse effects after their application. Likewise, to carry out the most used MRTs (MI), only the instructions of a trained professional are needed.

### 4.2. About the Population

The characteristics of MRT postulate it as a complementary option to conventional physical therapy (CPT), which could benefit those hospitalized patients with pathologies that produce a high degree of disability and that cause pain and gait alterations. In support of this suggestion, other studies indicate that MRT can effectively relieve and improve the range of motion in chronic musculoskeletal pain conditions [38], improving balance and gait skills in people with chronic stroke [39]. However, the certainty of the evidence is very low for short-term benefits on gait speed, considering other research [40] On the other hand, it is important to mention that the criteria that could exclude a person from the practice of MRT in a situation of rehabilitation should be analyzed, such as: neurological disorders, Parkinson’s, multiple sclerosis and cancer [33], as well as having methods to verify the execution of MRT, as is the case of the metric parameters of the gaze as an indicator of participation in a motor imagery task by a patient [41].

### 4.3. About the Intervention Based on MRT

Studies present multiple definitions of the concept. For this review, we considered movement representation techniques as any therapy that uses the representation of movement, especially observation and/or imagination of normal movements without pain. These approaches can be combined with movement execution and sensory stimulation to facilitate pain-free movements of the affected limb. Interventions include mirror therapy (MT), motor imagery (MI), and observation of movement and/or action (AOT), which consider the actual observation of a movement in another person, virtually or employing a video [19]. In the present review, the three most common forms of manifestation were found, which were: motor imagery (6 studies), action observation (2 studies), augmented reality (1 study), and action observation plus motor imagery (1 study). Therefore, it can be observed that there are different forms of expression of MRT, but the most studied was Motor imagery, which is defined as the brain process of construction of a motor action without the actual execution [42]. Other research support that, to some extent, motor images involve the same neuronal substrate as a real movement together with greater activation in the supplementary motor area, the premotor cortex, and the primary motor cortex, as well as an executed movement [36]. Moreover, evidence in stroke patients has shown that left lesions affect motor imagery in both hands, especially the speed of the executed movement; in contrast, right lesions affect the performance in the left hand [43]. Furthermore, this technique closely resembles the realization of the actual movement in the functioning of the nervous system. On the other hand, the studies present multiple methods of application of the MRT and different results for the same variable in some cases. Additionally, it would be appropriate to generate a standardized protocol [44] application of the MRT due to their need to be more information on the technique, the training prescription framework, and the eligibility criteria for the type of participants [45]. In addition, there are conflicting findings in motor imagery assessments [46], which could point out the need to implement valid assessments of these capabilities. Concerning to the variables that affect the application and effects of MRT, the capacity of previous motor imagination can enhance the results of the application of the techniques; this can occur due to the corticospinal excitability that increases, to a greater extent, in people with greater ability to generate motor images than those with less ability [47]. It is of further importance to consider that the corticospinal effects of a simple motor visualization task can predict the corticospinal effects of a more complex motor visualization task, involving the same muscle [48].

Regarding other aspects, pain caused by musculoskeletal conditions can affect the ability to generate mental representations of movement, with greater consistency in the pain of peripheral than in axial body segments [49,50]. In the present research, the variable pain presented 71% of statistically significant improvements, which may indicate a favorable outcome for using MRT. Additionally, note that mental images can be interrupted in patients with knee osteoarthritis [51]. Observation of the action has shown possible beneficial effects in the restart of motor activation, markedly reduced, in patients with fibromyalgia [52].

In the selected studies, the use of MRT is conducted in a complementary way, not as a single treatment or as an alternative to enhance therapeutic processes. MRT is a safe therapeutic tool that could be promoted in acute stages when people’s physical activity is reduced. Similarly, MRT is economical and easy to apply following an established protocol. Likewise, it is a common tool in physiotherapy, which could also be considered for other surgeries, extending the range of use. It is common in neurology, but in other areas, such as traumatology, it has been scarcely studied to explore its application and results [53].

### 4.4. The Effects of Movement Representation Techniques

MRT has no harmful effects in addition to conventional physical therapy. It is suggested as an adjunct to conventional therapy; combined with usual treatments, it positively impacts improving function [54]. It is also an intervention with a low cost [45], no further infrastructure is necessary, and it can be done remotely or at home [33,55]. Due to the characteristics and effects of the MRT intervention, it can be a good choice to attenuate the loss of muscle strength and gait alterations in periods of immobilization, in which physical and mental activation can be performed without the need for open movements [56], Movement representation techniques allow to increase corticomotor excitability and modulate intracortical inhibition [47]. Other studies on AOT have observed effects on corticospinal excitability causing early and nonspecific facilitation of this motor pathway (about 90 ms from the onset of action), followed by subsequent modulation of the specific activity of the muscles involved in the observed action (from about 200 ms) [57]. Concerning to MI, there is evidence of the existence of two types, kinesthetic images, and visual images, distinguished by the activation and inhibition of different brain areas related to motor α and β, and frequency regions. Brain activity corresponding to MI is generally observed in specially trained subjects or athletes; it is also possible to identify characteristics of MI in untrained subjects [58]. MRTs may also be useful in other areas of human performance, for example, during periods of forced detraining in professional athletes; the practice of MI appears to be a viable tool for maintaining and increasing physical performance capacity [59].

This review examines the available evidence on performing MRT at home and presents the most popular cognitive techniques used to increase physical performance, image visualization practices, and motor action. MI and different modalities of MRTs can be efficiently integrated into rehabilitation practice for orthopedic patients [56]. Additionally, Home motor imagery intervention improves functional performance after short-term total knee arthroplasty [33].

### 4.5. What Are the Limitations of This Review and the Review Processes Used?

It is an incipient area of study in Latin America, and recent articles are mainly concentrated in Europe. The highest risk of bias reported by the Cochrane tool was related to performance bias and detection bias. Both biases are associated with difficulty in masking participants and study staff. These biases increase the risk that the knowledge of the intervention received will affect the results, which could lead to a systematic error in the application of MRTs. It is relevant to highlight that numerous investigations indicate the low methodological quality of the studies [28]; other studies mention a high risk of bias in at least one domain evaluated [40]. In the population that uses MI in people who had a vascular accident, there is high heterogeneity in the methodological quality of the studies and contradictory results [44,60]. Another limitation of the review is that a qualitative analysis of the evidence we found was carried out only; a meta-analysis was not included. On the other hand, it is possible that new articles could be published after the search ended in October 2021, so the search was carried out again before the publication and two new articles were found that meet the inclusion criteria [61,62].

### 4.6. Contributions, Clinical Implications, and Future Lines of Research

To our knowledge, this systematic review is the first to look at the effects of MRT intervention combined with CPT compared with conventional physical therapy as a single intervention for patients with hip and knee arthroplasty. The evidence provided in this systematic review will clarify the characteristics of MRT in people undergoing knee and hip arthroplasty, as well as its main effects on variables such as physical health, functionality, cognitive function, and quality of life. In addition, it will help to know the main forms of application of MRT used in research and thus lay a basis for future designs or implementations of this technique. Furthermore, this review allowed us to detect the existence of gaps in knowledge regarding the standardization of participants or suitable instruments for evaluation, which may encourage the development of future studies in this area.

## 5. Conclusions

The literature indicates that MRTs are mainly performed as motor imagery and developed with heterogeneous frequencies and duration. MRT is a safe and low-cost therapeutic intervention. MRT combined with CPT generates beneficial effects on pain control, muscle strength gain, ROM, speed, gait, and functional capacity, as well as on motor visualization capacity, cognitive performance, and quality of life. When MRT combined with CPT is compared with CPT alone, it is observed that MRT combined with CFT has greater benefits in all physical, functional, cognitive, and quality of life variables analyzed. More studies of high methodological quality are needed to confirm the findings of this review.

## Figures and Tables

**Figure 1 sports-10-00198-f001:**
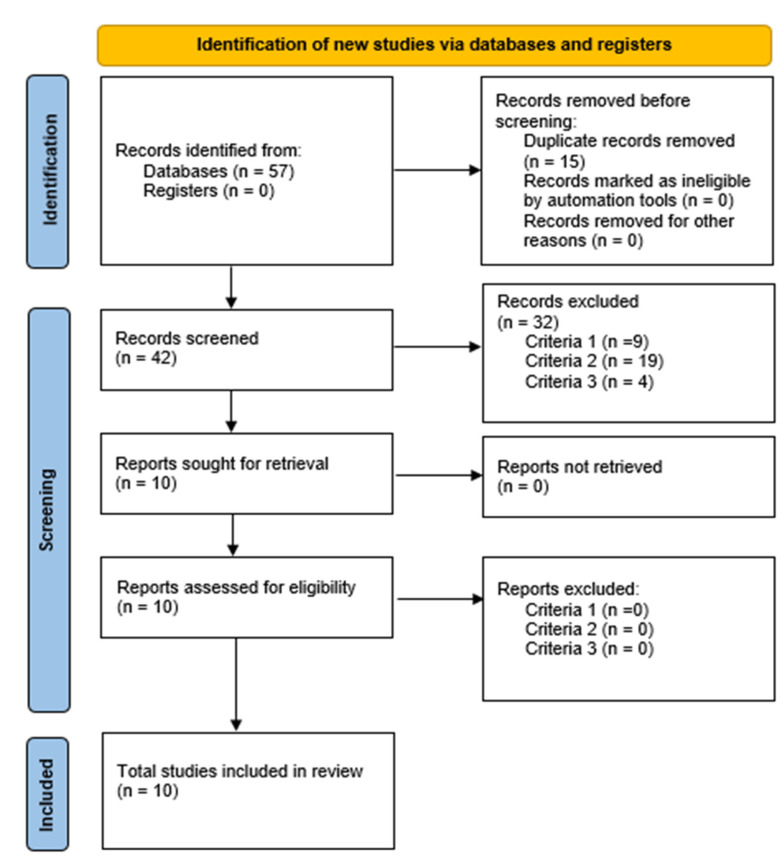
PRISMA flowchart.

**Figure 2 sports-10-00198-f002:**
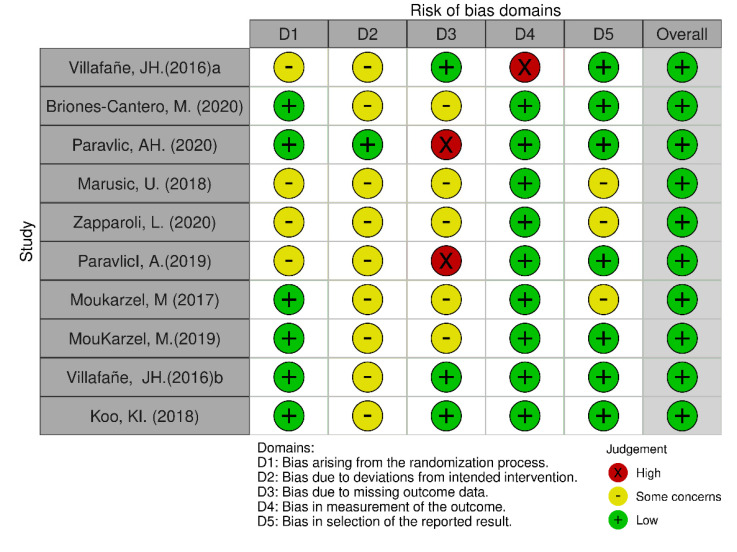
Semaphore diagram of the risk of bias of the selected articles [6,25,26,31,32,33,34,35,36,37].

**Figure 3 sports-10-00198-f003:**
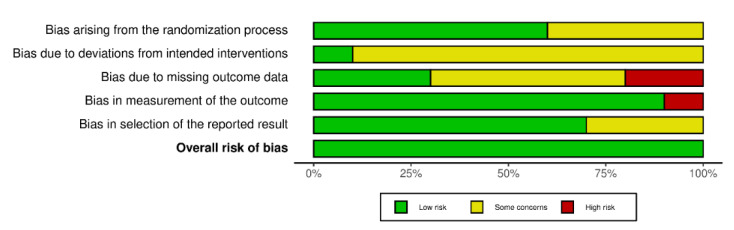
Synthesis of risk of bias.

**Table 1 sports-10-00198-t001:** Selection criteria.

Criterion	Description
(1) Population	(a) Use studies about men or women undergoing total or partial hip or knee arthroplasty.
(2) Intervention	Containing interventions of motion representation techniques such as motor imagery, grade motor imagery, action observation, mirror therapy, and movement representation techniques [19].
(3) Comparison	(a) Conventional physical rehabilitation; therapeutic exercise, manual therapy, physical agents, and education. (b) Conventional physical rehabilitation plus Placebo.
(4) Results	Valuing the following: (a) Physical variables. (b) Functionality variables. (c) Variables of cognitive function and quality of life
(5) Type of article	(a) Original experimental analytical articles randomized and non-randomized clinical trials. (b) Publications until October 2021. (c) No language restriction.

**Table 2 sports-10-00198-t002:** Main characteristics of the selected studies.

Author	Year	Country	Population (Age m ± SD)	Sex (M%)	Sex (F%)	CS	Intervention	Variables	Type of Design
Marusic, U. [6]	2018	Slovenia	21 (EXP 64.4 ± 4.1; CON 63.1 ± 5.6)	14 (66.7%)	7 (33.3%)	THA	AOT + MI	Physical function and cognitive function	RCT
Zapparoli, L. [25]	2020	Italy	48 (66.4 ± 7.7)	20 (41.7%)	28 (58.3%)	TNA	MI	functionality, knee ROM, knee pain intensity, gait analysis and risk of falls	RTC with placebo
ParavlicI, A. [26]	2019	Slovenia	34 (61.1 ± 5.3)	19 (55.9%)	15 (44.1%)	TNA	MI	physical function, spatiotemporal gait parameters, reported author physical function, and cognitive performance	RCT of parallel groups
Villafañe, J.H. [31]	2016	Italy	24 (69 ± 8.5)	10 (41.7%)	14 (58.3%)	THA	AOT	Hip pain intensity, hip ROM, functionality, and quality of life	PCE
Briones-Cantero, M. [32]	2020	Spain	24 (EXP 73 ± 5; CON 72 ± 6)	15 (62.5%)	9 (37.5%)	TNA	MI	pain-related disability, knee pain intensity, knee ROM, pain sensitivity to pressure	RCT of parallel groups
Paravlic, A.H. [33]	2020	Slovenia	26 (EXP 61.69 ± 5.19; CON 58.85 ± 5.24)	14 (53.8%)	12 (46.2%)	TNA	MI	maximum voluntary isometric strength of knee extension, voluntary activation of knee extension, functionality, ROM knee, intensity of knee pain, maximum grip strength and reported author function	RCT of parallel groups
Moukarzel, M. [34]	2017	France	20 (69.60 ± 3.25)	4 (20%)	16 (80%)	TNA	MI	knee pain, knee rom, knee circumference, quadriceps strength and functionality	RCT
MouKarzel, M. [35]	2019	France	24 (70 ± 2.89)	4 (16.7%)	20 (83.3%)	TNA	MI	ipsilateral quadriceps strength, maximum knee flexion during rocking phase and functionality	RCT
Villafañe, J.H. [36]	2016	Italy	31 (EXP 70.4 ± 7.5; CON 70.1 ± 7.7)	10 (32.3%)	21 (67.7%)	TNA	AOT	Knee pain intensity, quality of life, function, and gait	RCT Pilot
Koo, K.L. [37]	2018	South Korea	42 (FULL 65.00 ± 6.97; HALF 63.71 ± 5.09)	10 (23.8%)	32 (76.2%)	TNA	AR	knee pain at rest and activity, ROM knee	RCT, prospective, parallel group
			Total *n* = 294	*n* = 120 (40.8%)	*n* = 174 (59.2%)				

Caption: THA: Total hip arthroplasty, TNA: Total knee arthroplasty, AOT: Action observation therapy, MI: Motor imagery, AR: augmented reality. PCE: Prospective clinical study, RCT: Randomized clinical trial.

**Table 3 sports-10-00198-t003:** Interventions characteristics.

Author [Reference]	Groups	Frequency	Treatment							Extension	Duration
			MRT			CPT					
		Days/Week	AOT	MI	ER	AM	PM	TP	WI	Weeks	Minutes
Marusic, U. [6]	Control	3	–	–	–	NS	NS	NS	NS	9	NS
	Intervention	3	✓	✓	–	NS	NS	NS	NS	9	NS
Zapparoli, L. [25]	Control	6	–	–	–	✓	✓	✓	✓	2	70
	Intervention	6	–	✓	–	✓	✓	✓	✓	2	NS
Paravlic, A.H. [26]	Control	5	–	–	–	✓	✓	✓	✓	5	NS
	Intervention	5	–	✓	–	✓	✓	✓	✓	5	NS
Villafañe, J.H. [31]	Control	5	–	–	–	✓	✓	✓	✓	2	15
	Intervention	5	✓	–	–	✓	✓	✓	✓	2	30
Briones-Cantero, M. [32]	Control	5	–	–	–	✓	✓	✓	–	1	30
	Intervention	5	–	✓	–	✓	✓	✓	–	1	30
Paravlic, A.H. [33]	Control	5	–	–	–	✓	✓	✓	✓	4	NS
	Intervention	5	–	✓	–	✓	✓	✓	✓	4	45–60
Moukarzel, M. [34]	Control	3	–	–	–	✓	✓	✓	✓	4	60
	Intervention	3	–	✓	–	✓	✓	✓	✓	4	75
Moukarzel, M. [35]	Control	3	–	–	–	✓	✓	✓	✓	4	60
	Intervention	3	–	✓	–	✓	✓	✓	✓	4	60
Villafañe, J.H. [36]	Control	5	–	–	–	✓	✓	✓	✓	2	100
	Intervention	5	✓	–	–	✓	✓	✓	✓	2	100
Koo, K.L. [37]	HTI	5	–	–	–	NS	✓	NS	✓	1	30
	FTI	5	–	–	✓	NS	✓	NS	✓	2	30

Caption: AOT: action observation therapy, MI: motor imagery, ER: enhanced reality, AM: active mobilization, PM: passive mobilization, TP: transfer practice, ✓: reported information, NS: not specified, FTI: full term intervention, HFI: half term intervention.

**Table 4 sports-10-00198-t004:** Effect of movement representation techniques.

Ref.	Groups	Physical	Functionality	Cognitive	Quality of Life	Other
		Pain	Strength	Rom	Speed	Gait	Functional Capacity	Motor Visualization Capability	Cognitive Performance		
[6]	Co					↓TUG, ↓FSST, ↓S&D-TW(GS), ↓S&D-TW(OTV)			 S&D-TW(CP),  S&D-CP(CP)		
In					 *TUG, ↑*FSST,  S&D-TW(GS),  *S&D-TW(OTV)			↑*S&D-TW(CP), ↑S&D-CP(CP)		
[25]	Co	 VAS		↑P-ROM		↑TUG, ↑GTS, ↑AR-GET	↑FIS, ↑BARTHEL, FNF	 MIQI,  HWT			
In	↑*VAS		↑P-ROM		↑*TUG, ↑GTS, ↑*AR-GET	↑FIS, ↑BARTHEL, *FNF	↑*MIQI,  HWT			
[26]	Co		↑ MIEK, ↑SSC		↑SWS, ↑SRSDT, ↑BWS, ↑DTBWS	↑SSP,  DSP, *↑SL, *↑cadencia	 LEFS	↑KIC,  EVIC, ↑IVIC	↑baseline ↑SSWS, ↑SMR		
In		*↑ MIEK, *↑SSC		*↑SWS, *↑SRSDT, *↑BWS, *↑DTBWS	*↑SSP, *  DSP, ↑SL, ↑cadencia	 LEFS	*↑KIC,  EVIC,*↑IVIC	↑baseline, ↑SSWS, ↑SMR		
[31]	Co	↑VAS		↑A-ROM-HA, ↑P-ROM-HA			↑TINETTI, ↑LEQUESNE, BARTHEL (N.D)			↑SF-36 PF,  SF-36MHF	
[32]	Co	↑VAS,  PPT		 KFKE-ROM			↑WOMAC				
[33]	Co	↑OL-VAS, ↑NOL-VAS	↓ MIEK, ↓VA-OL,  MVIS-UL,  VA-UL,  DAR	↑OL-KF, ↑OL-KE,  NOL-KF,  NOL-KE		↓TUG	 OKS				
[34]	Co	↑VAS	↑ MIEK	 A-ROM,  P-ROM		↑TUG					 KC
In	*↑VAS	*↑ MIEK	↑A-ROM, ↑P-ROM		↑TUG					 KC
[35]	Co		↑ MIEK	↑MKFR		↑TUG, ↑6MWT	↑OKS, ↑SCT				
In		↑ MIEK	*↑MKFR		↑TUG, ↑6MWT	↑OKS, *↑SCT				
[36]	Co	↑VAS		↑A-ROM, ↑P-ROM			↑TINETTI, ↑LEQUESNE, ↑BARTHEL			↑SF-36 PF,  SF-36MHF, CDRS (ND)	
In	↑VAS		↑A-ROM, ↑P-ROM			↑TINETTI, ↑LEQUESNE, ↑BARTHEL			↑SF-36 PF,  SF-36MHF, CDRS (NS)	
[37]	*Co	↑VAS	↑SSC	↑A-ROM		↑GAD, ↑6MWT	↑WOMAC			 GDSSF	
In	*↑VAS	↑SSC	*↑A-ROM		↑GAD, ↑6MWT	↑WOMAC			 GDSSF	

Caption: ↑: Statistically significant improvement compared to the initial assessment, *: Statistically significant improvement compared to the control group, ↓: Statistically significant decrease compared to the initial assessment, ↔: No statistically significant changes compared to the initial assessment, Co: control, In: Intervention INS: Not Specified, MIEK: Maximum isometric extension force of the intervened knee (Nm), SSC: Sit down and stand up from the chair for 30 s (*n*), SWS: Self-selected walking speed (m/s), SRSDT: Self-selected running speed DT (m/s), BWS: Brisk walking speed (m/s), DTBWS: DT brisk walking speed (m/s), SSP: Single Support Period(s), DSP: Dual Support Period(s), SL: Stride length, LEFS: Lower Extremity Functional Scoring Questionnaire, KIC: Kinesthetic imagery capacity, EVIC: External visual imaging capability, IVIC: internal visual imaging capability, SSWS: Series 3s during self-selected walking speed, SMR: Series 3 during fast walking, TUG: Time Up and Go, KC: Knee circumference, MKFR: Maximum knee flexion during the rocking phase, 6MWT: 6 min walk test, SCT: Stair climbing test, OKS: Oxford knee score, CDRS: Cumulative Disease Rating Scale, GDSSF: Geriatric Depression Scale Short Form, GAD: Graduated ambulation distance, NOL-VAS: Non-operated leg-analog visual scale, VA-OL: Voluntary Muscle Activation of the Operated Leg, MVIS-UL: Maximum Voluntary Isometric Strength of the Unoperated Leg, VA-UL: Voluntary Muscle Activation of the Unoperated Leg, DAR: Dominant Arm Handgrip, A-ROM-HA: Active ROM of Hip Abduction and Flexion, P-ROM-HA: Pasive ROM of Hip Abduction and Flexion KFKE-ROM: Knee Flexion and Extension ROM, SF-36 PF: SF-36 Physical Functional, SF-36 MHF: SF-36 Mental Health Functional, PPT: Pressure Pain Threshold, WOMAC: McMaster University of Western Ontario Osteoarthritis Index, S&D-TW(GS): Single and Dual task walking-Gait Speed, S&D-TW(OTV): Single and Dual task walking-Oscillation Time Variability, S&D-TW(CP):Single and Dual task walking-Cognitive Performance, S&DT-CP(CP): Single and Dual task Control Postural-Cognitive Performance, GTS: Gait Temporal Space Variables, AR-GET: Active ROM of Gait Space-Time Variables, P-ROM: Passive ROM, FIS: Functional Independence Scale, FNF: Falls and Near Falls, MIQI: Moto Imagery Quality Index TUG, HWT: Hang Walking task.

## Data Availability

Data will be made available upon request.

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
