# Peer review of "Effects of the Practice of Movement Representation Techniques in People Undergoing Knee and Hip Arthroplasty: A Systematic Review"

_sports, 2022, doi:10.3390/sports10120198_

Round 1
Reviewer 1 Report
The subject of study seems very interesting to me, however there are some aspects to modify and take into account.
First of all, the abstract of the article is too long, the results section should be better summarized.
Why is your search carried out a year before sending it to publish? Have new articles come out that could be included? This would be very important since it could modify the data
the databases where the search for this article is focused are too scarce. I recommend redoing it and looking at SCOPUS, Web of Science, Cinahl, PEDro...
In the results, what are the exclusion criteria that have been carried out in each of the exclusions?
Is there the possibility of combining the results of the study variables because a meta-analysis has not been carried out to provide something different?
Table 4 has a poor image quality, it would be advisable to change it and improve it.
Author Response
The subject of study seems very interesting to me, however there are some aspects to modify and take into account.
Comment 1: First of all, the abstract of the article is too long, the results section should be better summarized.
Answer 1: the abstract was shortened.
Comment 2: Why is your search carried out a year before sending it to publish? Have new articles come out that could be included? This would be very important since it could modify the data
Answer 2: Thank you very much for the comment. It is true that a year has passed since the search ended, although from my experience carrying out systematic reviews, it is a reasonable period.
The main arguments that explain this time are: (i) enrollment in PROSPERO took about 4 months, (ii) all other actions after the search took 6 months (screening and inclusion of articles, writing the review and review by all authors).
The authors are aware that new articles could be published from the search, so this situation is indicated in the limitations section.
In this line, the search was carried out again and two new articles were found that were added in the limitations section of the review.
- 10.1177/02692155221116820
- 10.3390/medicina58070868
Comment 3: The databases where the search for this article is focused are too scarce. I recommend redoing it and looking at SCOPUS, Web of Science, Cinahl, PEDro...
Answer 3: We believe that the chosen databases cover a high percentage of the articles that studied the effects of movement representation techniques in people undergoing knee and hip arthroplasty.
On the other hand, Web of Science was used and if indicated in line 99.
CINHAL was not considered, since it is a database focused on nursing articles, and PEDro was not considered either, since it has a low number of articles.
In any case, today we searched the databases: CINHAL and PEDro and no new articles were found.
Attached is the CINAHL search syntax (motor imagery OR action observation OR graded motor imagery OR mirror therapy OR movement representation techniques) AND (S1 AND S2) and the PEDro search link is attached: https://search.pedro.org.au/advanced-search/results?abstract_with_title=motor+imagery&therapy=0&problem=0&body_part=VL01399&subdiscipline=0&topic=0&method=0&authors_association=&title=arthroplasty&source=&year_of_publication=&date_record_was_created=&nscore=&perpage=20&lop=and&find=&find=Start+Search
Comment 4: In the results, what are the exclusion criteria that have been carried out in each of the exclusions?
Answer 4: exclusion criteria were exposed in the selection of studies item, line 112-116 “We included randomized and non-randomized clinical trials, excluding systematic reviews, quasi-experimental studies, observational studies, editorial papers, conference proceedings, protocols, and theses. Articles selected by title, abstract and full text had to meet the conditions indicated in Table 1”.
Comment 5: Is there the possibility of combining the results of the study variables because a meta-analysis has not been carried out to provide something different?
Answer 5: We appreciate the comment, and it would be very interesting to do a meta-analysis in the next study. As commented in the review limitations section (lines 397-398), a qualitative analysis was performed without including a quantitative analysis.
Comment 6: Table 4 has a poor image quality; it would be advisable to change it and improve it.
Answer 6: to improve the quality of table 4, the font size is enlarged.
Reviewer 2 Report
1. The introduction is very well grounded, but please emphasise the elements of originality/ novelty that this study brings to the specialised literature. I understand that this review aims to analyse the effects of movement representation techniques combined with conventional physical therapy in people undergoing knee and hip arthroplasty compared to conventional physical therapy alone in terms of results in physical and functionality variables, cognitive function, and quality of life, so please:
· Please insert the novelties that this study brings at the end of the introduction;
· Please better highlight the purpose of this study.
2. Please change Cumming, J.; Ramsey, R. Imagery Interventions in Sport. 2009, doi:10.13140/2.1.2619.2322. with Cumming, J.; Ramsey, R. Imagery Interventions in Sport. In Mellalieu, S., & Hanton, S. (Eds.). Advances in Applied Sport Psychology: A Review (1st ed.). Routledge. 2008, https://doi.org/10.4324/9780203887073.
3. We identified all the articles published in the databases that studied the effects of movement representation techniques in people undergoing knee and hip arthroplasty (lines 101-103). How did you identify the items? Was the reference manager tool EndNote used? Do you use Zotero program for that too, not just for removing duplicates? Please specify.
4. Lines 174-176. Please enter on caption ECA and RCT (or change with RCT)
5. The article has been entered into the software Plagiarism Checker X and has no problem with the similarity coefficients.
6. I congratulate the authors for their work in producing this systematic review!
Author Response
Comment 1. The introduction is very well grounded, but please emphasise the elements of originality/ novelty that this study brings to the specialised literature. I understand that this review aims to analyse the effects of movement representation techniques combined with conventional physical therapy in people undergoing knee and hip arthroplasty compared to conventional physical therapy alone in terms of results in physical and functionality variables, cognitive function, and quality of life, so please:
- Please insert the novelties that this study brings at the end of the introduction;
- Please better highlight the purpose of this study.
Answer 1: Thank you very much for the comment, information is added in the introduction that accounts for the originality and novelty of the systematic review.
Comment 2. Please change Cumming, J.; Ramsey, R. Imagery Interventions in Sport. 2009, doi:10.13140/2.1.2619.2322. with Cumming, J.; Ramsey, R. Imagery Interventions in Sport. In Mellalieu, S., & Hanton, S. (Eds.). Advances in Applied Sport Psychology: A Review (1st ed.). Routledge. 2008, https://doi.org/10.4324/9780203887073
Answer 2: Thank you very much for the comment, the reference was changed.
Comment 3. We identified all the articles published in the databases that studied the effects of movement representation techniques in people undergoing knee and hip arthroplasty (lines 101-103). How did you identify the items? Was the reference manager tool EndNote used? Do you use Zotero program for that too, not just for removing duplicates? Please specify.
Answer 3: Thanks for the comment, it clarified how the articles were identified.
Comment 4. Lines 174-176. Please enter on caption ECA and RCT (or change with RCT)
Answer 4: Thanks for the comment, the error was corrected in the table 2.
Comment 5. The article has been entered into the software Plagiarism Checker X and has no problem with the similarity coefficients.
Answer 5: Excellent
Comment 6. I congratulate the authors for their work in producing this systematic review!
Answer 6: Thank you! very much
Round 2
Reviewer 1 Report
the authors have responded correctly to the comments